# Adaptive Multi-stage Density Ratio Estimation for Learning Latent Space Energy-based Model

**Zhisheng Xiao**
University of Chicago
Chicago, IL, 60637
zxiao@uchicago.edu

**Tian Han**
Stevens Institute of Technology
Hoboken, NJ, 07030
than6@stevens.edu

## Abstract

This paper studies the fundamental problem of learning energy-based model (EBM) in the latent space of the generator model. Learning such prior model typically requires running costly Markov Chain Monte Carlo (MCMC). Instead, we propose to use noise contrastive estimation (NCE) to discriminatively learn the EBM through density ratio estimation between the latent prior density and latent posterior density. However, the NCE typically fails to accurately estimate such density ratio given large gap between two densities. To effectively tackle this issue and learn more expressive prior models, we develop the adaptive multi-stage density ratio estimation which breaks the estimation into multiple stages and learn different stages of density ratio sequentially and adaptively. The latent prior model can be gradually learned using ratio estimated in previous stage so that the final latent space EBM prior can be naturally formed by product of ratios in different stages. The proposed method enables informative and much sharper prior than existing baselines, and can be trained efficiently. Our experiments demonstrate strong performances in image generation and reconstruction as well as anomaly detection.

## 1 Introduction

Deep generative model provides a powerful framework for representing complex data distributions and have seen many successful applications in image and video synthesis [21, 43, 51], representation learning [34] as well as unsupervised or semi-supervised learning [19, 37]. Such model, referred to as *generator model*, usually consists of low-dimensional latent variables that follow non-informative prior distribution, and a top-down network that maps such latent vector to the observed example. The informative prior model in the latent space [36, 1] can be learned to further improve the expressive power of the whole model. Specifically, we consider learning energy-based model (EBM) in the latent space as our informative prior for the generator model.

Learning latent space EBM can be challenging and requires iterative Markov Chain Monte Carlo (MCMC) sampling step which is computationally expensive and sensitive to hyperparameters. In this paper, we instead propose to use noise contrastive estimation (NCE) [12] for learning EBM prior via density ratio estimation. The EBM is learned discriminatively by classifying the latent vector sampled from the prior density and the latent sampled from posterior density. Instead of variational learned inference [23, 41] which needs a separate inference network designed, we obtain the posterior latent sample through short-run Langevin dynamics [33] to ensure more accurate inference. However, the success of NCE depends on the closeness of prior density and posterior density [18]. Given large gap between two densities, NCE typically fails to accurately estimate such density ratio which leads to inaccurate EBM modeling.

To effectively tackle the inaccurate estimation issue and further learn more expressive prior model, we develop the adaptive multi-stage density ratio estimation for latent space EBM training. The proposed

36th Conference on Neural Information Processing Systems (NeurIPS 2022).

model breaks the density estimation into multiple stages and learn different stages of density ratio sequentially. Thanks to the low-dimensionality of the latent space and the short-run style posterior inference, in each stage, the gap between prior and posterior density could be kept in check which makes NCE easier. The density ratio estimated in previous stage can be further integrated into the current prior model as a correction term to build more expressive prior density for later stage. With such framework, the final latent space EBM prior can then be naturally formed by product of ratios in different stages on top of the initial base prior.

**Contributions:** 1) we propose an EBM prior on generator model which is modelled through estimation of density ratios in multiple stages. 2) We develop the adaptive multi-stage noise contrastive estimation to learn different stages of ratios sequentially and adaptively. The ratio estimated in previous stage can be integrated to form the more informative prior in the later stage. 3) we demonstrate strong empirical results to illustrate the proposed method.

## 2 Background

### 2.1 Maximum likelihood learning of deep latent variable models

Let $\mathbf{x} \in \mathbb{R}^D$ be an observed example such as an image, and $\mathbf{z} \in \mathbb{R}^d$ be the latent variables where $d < D$. A latent variable generative model (a.k.a, *generator model*) factorize the joint distribution of $(\mathbf{x}, \mathbf{z})$ as

$$p_\theta(\mathbf{x}, \mathbf{z}) = p(\mathbf{z})p_\theta(\mathbf{x}|\mathbf{z}), \tag{1}$$

where $p(\mathbf{z})$ is the prior distribution over latent variables $z$, $p_\theta(\mathbf{x}|\mathbf{z})$ is the top-down generation model with parameters $\theta$. Usually the prior distribution is chosen to be a simple one such as $\mathcal{N}(0, \mathbf{I}_d)$, but it can also be more expressive with learnable parameters [36]. The generation model is the same as that in VAE [23], i.e., $\mathbf{x} = g_\theta(\mathbf{z}) + \epsilon$ with $g_\theta$ to be the decoder network and $\epsilon \sim \mathcal{N}(0, \sigma^2\mathbf{I}_D)$, so that $p_\theta(\mathbf{x}|\mathbf{z}) = \mathcal{N}(g_\theta(\mathbf{z}), \sigma^2\mathbf{I}_D)$. As in VAE, $\sigma^2$ takes a pre-specified value.

Given a set of $N$ training samples $\{\mathbf{x}_i, i = 1, \ldots, N\}$ from the unknown data distribution $p_{\text{data}}(\mathbf{x})$, the model $p_\theta$ can be trained by maximizing the log likelihood over training samples $\mathcal{L}(\theta) = \frac{1}{N}\sum_{i=1}^N \log p_\theta(\mathbf{x}_i)$. Maximizing the log likelihood $\mathcal{L}(\theta)$ can be accomplished by gradient ascent where the gradient can be obtained from

$$\nabla_\theta \log p_\theta(\mathbf{x}) = \frac{1}{p_\theta(\mathbf{x})}\nabla_\theta p_\theta(\mathbf{x}) = \int \left[\nabla_\theta \log p_\theta(\mathbf{x}, \mathbf{z})\right] \frac{p_\theta(\mathbf{x}, \mathbf{z})}{p_\theta(\mathbf{x})} d\mathbf{z}$$
$$= \mathbb{E}_{p_\theta(\mathbf{z}|\mathbf{x})}\left[\nabla_\theta \log p_\theta(\mathbf{x}, \mathbf{z})\right]. \tag{2}$$

$\nabla_\theta \log p_\theta(\mathbf{x}, \mathbf{z})$ can be easily computed according to the form of $\log p_\theta(\mathbf{x}, \mathbf{z})$, however, approximating the expectation requires drawing samples from $p_\theta(\mathbf{z}|\mathbf{x})$, which can be difficult. Sampling from the intractable posterior $p_\theta(\mathbf{z}|\mathbf{x})$ requires MCMC, and one convenient MCMC algorithm is Langevin Dynamics (LD) [30]. Given a step size $s > 0$, and an initial value $\mathbf{z}_0$, the Lanegvin dynamics iterates

$$\mathbf{z}^{k+1} = \mathbf{z}^k + \frac{s}{2}\nabla_\mathbf{z} \log p_\theta(\mathbf{z}|\mathbf{x}) + \sqrt{s}\omega_k, \tag{3}$$

where $\omega_k \sim \mathcal{N}(0, \mathbf{I})$. For sufficiently small step size $s$, the marginal distribution of $\mathbf{z}_k$ will converge to $p_\theta(\mathbf{z}|\mathbf{x})$ as $k \to \infty$. However, it is not feasible to run Langevin dynamics until convergence, and in practice the iteration in Eq. 3 is run for finite iterations, which yields a Markov chain with an invariant distribution approximately close to the original target distribution. When $\mathbf{z}_0$ is initialized from the noise distribution, the algorithm is called noise-initialized short-run LD [32, 33].

### 2.2 Learning EBMs with discriminative density ratio estimation

Suppose there are two distributions with density functions $p(\mathbf{x})$ and $q(\mathbf{x})$ from which we can sample, we can estimate the density ratio[1] $r(\mathbf{x}) = \frac{p(\mathbf{x})}{q(\mathbf{x})}$ by training a classifier to distinguish samples from $p$ and $q$ [44]. Specifically, we can train the binary classifier $D : \mathbb{R}^n \to (0, 1)$ by minimizing the binary cross-entropy loss

$$\min_D -\mathbb{E}_{\mathbf{x}\sim q(\mathbf{x})}[\log D(\mathbf{x})] - \mathbb{E}_{\mathbf{x}\sim p(\mathbf{x})}[\log(1 - D(\mathbf{x}))].$$

---

[1]Assuming $q(\mathbf{x}) > 0$ when $p(\mathbf{x}) > 0$.

The objective is minimized when $D(\mathbf{x}) = \frac{q(\mathbf{x})}{q(\mathbf{x})+p(\mathbf{x})}$ [11], and denoting the classifier at optimality by $D^*(\mathbf{x})$, we have $r(\mathbf{x}) = \frac{p(\mathbf{x})}{q(\mathbf{x})} \approx \frac{1-D^*(\mathbf{x})}{D^*(\mathbf{x})}$. Equivalently, the ratio $r(\mathbf{x}) = \frac{p(\mathbf{x})}{q(\mathbf{x})}$ can be estimated by directly minimizing

$$\mathcal{L}(\phi) = -\mathbb{E}_{\mathbf{x} \sim p(\mathbf{x})} \log \left( \frac{r_\phi(\mathbf{x})}{1 + r_\phi(\mathbf{x})} \right) - \mathbb{E}_{\mathbf{x} \sim q(\mathbf{x})} \log \left( \frac{1}{1 + r_\phi(\mathbf{x})} \right), \tag{4}$$

where $r_\phi(\mathbf{x})$ is a non-negative ratio estimating model implemented as the exponential of an unconstrained neural network with scalar output. The minimizer $\phi^*$ satisfies $r_{\phi^*}(\mathbf{x}) = \frac{p(\mathbf{x})}{q(\mathbf{x})}$ [12].

Such a technique can be useful for training Energy-based models (EBMs). Given samples from the true data distribution $p_{\text{data}}(\mathbf{x})$ and a base distribution $q(\mathbf{x})$ that we can sample from, we consider EBMs of the form $p_\phi(\mathbf{x}) = \frac{1}{Z} r_\phi(\mathbf{x}) q(\mathbf{x})$, where $Z$ is the normalizing constant and $r_\phi$ is an unconstrained positive function. With this parametrization, obviously the optimal $r_\phi$ equals the density-ratio $\frac{p_{\text{data}}(\mathbf{x})}{q(\mathbf{x})}$. In fact, if $r_\phi(\mathbf{x})$ is trained with density ratio estimation, the normalizing constant $Z$ is simply 1. Therefore, the problem of learning an EBM becomes the problem of estimating a density-ratio, which can be solved by discriminative density ratio estimation. Typically the base distribution $q(\mathbf{x})$ is chosen to be Gaussian, resulted in so-called noise contrastive estimation (NCE) [12].

Although NCE provides a promising way to train EBMs without running MCMC, the accuracy of the density ratio estimation depends on the closeness between the two distributions. The ratio estimator is often severely inaccurate when the gap between $p$ and $q$ is large. Rhodes et al. [42] propose Telescoping density-Ratio Estimation (TRE), which breaks the density ratio estimation task into a collection of harder sub-tasks and show improvement over simple NCE on density ratio estimation. However, it is still difficult to apply the technique to energy-based modeling. On one hand, EBMs in high-dimensional data space such as image space can be highly complex and multimodal, making them extremely far away from simple noise distribution. On the other hand, the intermediate distributions are pre-designed through linear transition, making them less effective to connect complicated target densities. In Rhodes et al. [42], TRE only obtains limited success on training EBMs through density estimation on MNIST dataset.

## 3 Adaptive Multi-stage Desnity Ratio Estimation

In this section, we introduce adaptive multi-stage density ratio estimation on latent space in details.

### 3.1 Multi-stage density ratio estimation in latent space

Instead of modeling directly on high-dimensional data space, it is easier to introduce low-dimensional latent variables and learn an EBM in latent space, while also learning a mapping from the latent space to the data space [3, 26]. We follow this approach and attempt to model a latent space EBM using contrastive estimation.

The latent EBM can be learned discriminatively by estimating the ratio between prior density and the posterior density. Due to low-dimensionality of the latent space, such densities can be much easier to deal with than those in high dimensional data space. However, it presents new challenges. Firstly, while the target density in data space is given and fixed (i.e., empirical data distribution), posterior density in latent space is driven by the prior density and the inference on the posterior can be hard. Secondly, while the prior is typically assumed to be un-informative and fixed (e.g., unit Gaussian), the expressiveness of the model is limited.

Inspired by [42], we propose to learn the latent space EBM of the below form through multiple stages

$$p_\phi(\mathbf{z}) = \prod_{k=0}^{m-1} r_{\phi_k}(\mathbf{z}) \, p_0(\mathbf{z}), \tag{5}$$

where $p_0(\mathbf{z})$ is the unit Gaussian base distribution, and $r_{\phi_k}$ is the intermediate density ratio learned in each stage. Such proposed model shares the similar root as the Product-of-Expert (PoE) [17] where $r_{\phi_k}$ in each stage can be treated as individual expert model, and it has the potential to produce much sharper distribution than the one with single expert model built such as [36].

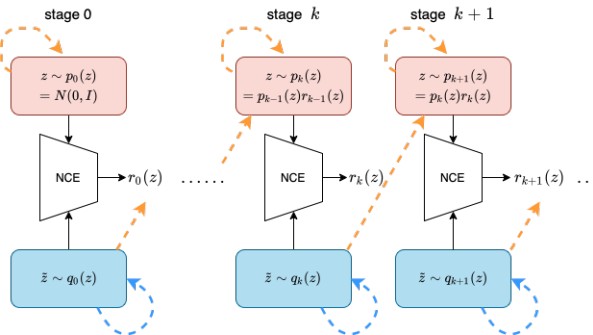

Figure 1: Training adaptive multi-stage density ratio estimation. We estimate the density ratio $r_k(\mathbf{z})$ in each stage using contrastive estimation which trains a classifier to distinguish samples from the prior $p_{\phi_k}(\mathbf{z})$ and samples from the aggregate posterior $q_k(\mathbf{z})$. Posterior samples are obtained by short-run LD (blue dashed curve), prior samples can be obtained either by short-run LD (orange dashed curve) or using persistent chain (orange dashed line). The ratio estimated in stage $k$ can be integrated to form a new prior in stage $k+1$. The whole prior is adapted across multiple stages and learned sequentially.

Although the formulation of EBM in Eq. 5 is related to the TRE proposed in Rhodes et al. [42], our training method is fundamentally different. In the next section, we will introduce the training of our model, and highlight the distinction with TRE.

### 3.2 Learning latent EBMs with adaptive multi-stage density ratio estimation

Our proposed generator model specifies the distribution on joint space $(\mathbf{x}, \mathbf{z})$: $p_{\theta,\phi}(\mathbf{x}, \mathbf{z}) = p_\phi(\mathbf{z})p_\theta(\mathbf{x}|\mathbf{z})$, where $p_\phi(z)$ is the prior model specified in Eq. 5, and $\phi = \{\phi_0, \ldots, \phi_{m-1}\}$ that collects parameters for all intermediate learned ratios.

It is tempting to apply maximum likelihood estimation (MLE) to train such model. However, there are several challenges: (1) learning of latent EBM $p_\phi(\mathbf{z})$ needs costly and hard mixing MCMC sampling. (2) the prior $p_\phi(\mathbf{z})$ needs to have a fixed form during training and cannot be adaptively adjusted. To alleviate the aforementioned limitations, we therefore break the density ratio estimation of $p_\phi(\mathbf{z})$ into $m$ stages, learn and build the prior sequentially and adaptively. Specifically, in the $k^{th}$ stage, we consider the generator model of the form

$$p_{\theta,\phi_k}(\mathbf{x}, \mathbf{z}) = p_{\phi_k}(\mathbf{z})p_\theta(\mathbf{x}|\mathbf{z}), \tag{6}$$

where $p_{\phi_k}(\mathbf{z}) = \prod_{i=0}^{k-1} r_{\phi_i}(\mathbf{z}) p_0(\mathbf{z})$. The whole training procedure iterates between the maximum likelihood estimation of generation model $\theta$ and the sequential contrastive estimation of prior $\phi$.

**MLE for generation model $\theta$:** The generation model can be trained by maximizing the marginal log-likelihood $p_\theta(\mathbf{x})$. In $k^{th}$ stage, the complete data log-likelihood of the model $p_{\theta,\phi_k}(\mathbf{x}, \mathbf{z})$ can be expressed as

$$\log p_{\theta,\phi_k}(\mathbf{x}, \mathbf{z}) = \log\left[p_{\phi_k}(\mathbf{z})p_\theta(\mathbf{x}|\mathbf{z})\right] = \log p_{\phi_k}(\mathbf{z}) - \frac{1}{2}\left[\|\mathbf{x} - g_\theta(\mathbf{z})\|^2 / \sigma^2\right] + C$$

where $g_\theta$ is the decoder and $C$ is a constant independent of $\theta$. The generation model parameter $\theta$ is then updated using the gradient based on Eq. 2 with a batch of training $n$ samples $\mathbf{x}_i$:

$$\theta_{t+1} = \theta_t + \eta_t \sum_{i=1}^{n} \mathbb{E}_{p_{\theta_t}(\mathbf{z}_i|\mathbf{x}_i)}\left[\left.\frac{\partial}{\partial\theta} \log p_{\theta,\phi_k}(\mathbf{x}_i, \mathbf{z}_i)\right|_{\theta=\theta_t}\right], \tag{7}$$

where $\eta_t$ is the learning rate. The expectation over the posterior can be approximated by running short-run Lanegvin dynamics in Eq. 3. Note that the running LD to sample from $p_\theta(\mathbf{z}|\mathbf{x})$ is equivalent to sample from $p_\theta(\mathbf{x}, \mathbf{z})$ with fixed $\mathbf{x}$.

**Adaptive multi-stage NCE for prior $\phi$:** The prior model $p_\phi(\mathbf{z})$ can be sequentially and adaptively learned to bridge the gap between prior and posterior densities in the previous stages. Specifically, in $k^{th}$ stage, the correction term $r_{\phi_k}$ can be trained to estimate the density ratio between $p_{\phi_k}(\mathbf{z})$ and its

aggregated posterior $q_k(\mathbf{z})$ through contrastive estimation using Eq. 4. The appealing advantage of this estimator is that it simply trains a binary classifier rather than using expensive MCMC sampling. The optimality of such logistic loss leads to the estimated $r_{\phi_k}(\mathbf{z}) \approx \frac{q_k(\mathbf{z})}{p_{\phi_k}(\mathbf{z})}$.

The prior model in the $(k+1)^{th}$ stage can then be sequentially adapted to match the previous aggregated posterior $q_k(\mathbf{z})$, i.e., letting $p_{\phi_{k+1}}(\mathbf{z}) = r_{\phi_k}(\mathbf{z})p_{\phi_k}(\mathbf{z})$ match $q_k(\mathbf{z})$. Given the new prior, we similarly infer the posterior using short-run LD in Eq. 3. Then, the next density ratio estimator $r_{\phi_{k+1}}$ can be learned through contrastive estimation to match the updated prior and its aggregated posterior, which is further used to adapt the prior in next stage. Particularly, we have

$$\frac{q_{m-1}(\mathbf{z})}{p_0(\mathbf{z})} = \frac{q_{m-1}(\mathbf{z})}{p_{\phi_{m-1}}(\mathbf{z})} \frac{q_{m-2}(\mathbf{z})}{p_{\phi_{m-2}}(\mathbf{z})} \cdots \frac{q_0(\mathbf{z})}{p_0(\mathbf{z})},$$

where $q_k(\mathbf{z})$ is the aggregated posterior for prior $p_{\phi_k}(\mathbf{z})$. The above telescoping product holds since the new prior is designed to match the aggregated posterior in previous stage, i.e., $p_{\phi_{k+1}}(\mathbf{z}) \approx q_k(\mathbf{z})$. Each stage estimates the ratio $r_{\phi_k}(\mathbf{z}) \approx \frac{q_k(\mathbf{z})}{p_{\phi_k}(\mathbf{z})}$ via contrasive estimation. Then the aggregated posterior $q_{m-1}(\mathbf{z})$ can be obtained via

$$q_{m-1}(\mathbf{z}) = r_{\phi_{m-1}}(\mathbf{z})\, p_{\phi_{m-1}}(\mathbf{z}) = \prod_{k=0}^{m-1} r_{\phi_k}(\mathbf{z})\, p_0(\mathbf{z}),$$

Our final prior model can then be obtained by matching such aggregated posterior $q_{m-1}(\mathbf{z})$ which has the same form as Eq. 5. The proposed training is illustrated in Figure 1 and the algorithm is detailed in Algorithm 1.

**Comparison with TRE in Rhodes et al. [42]:** The most significant difference between our training method and TRE is that TRE assumes a fixed target distribution and construct multiple stages simultaneously via interpolation, whereas our model considers adaptive targets and learn multi-stage density ratio estimators sequentially via NCE. We conduct empirical comparisons in Sec. 5.5.

**Sampling from prior $p_{\phi_k}(\mathbf{z})$:** The density ratio estimation of $r_{\phi_k}$ in each stage requires the samples from the prior $p_{\phi_k}(\mathbf{z})$ and posterior. The posterior samples are inferred through short-run Langevin dynamics which can be efficient and accurate. For drawing prior samples from $p_{\phi_k}(\mathbf{z})$, we can either use short-run prior Langevin dynamics or persistent update.

One one hand, we could directly utilize the short-run Langevin on the $p_{\phi_k}(\mathbf{z})$ to obtain prior samples. On the other hand, the samples from prior can also be obtained in a persistent chain manner to avoid the prior Langevin altogether. When introducing the $(k+1)^{th}$ stage of density ratio estimation, we assume that the current estimator $r_{\phi_k}(\mathbf{z})$ performs well in modeling the ratio between the current aggregated posterior distribution $q_k(\mathbf{z})$ and current prior $p_{\phi_k}(\mathbf{z})$. Therefore, we simply use samples from $q_k(\mathbf{z})$ to approximately serve as samples from the new prior $p_{\phi_{k+1}}(\mathbf{z})$ for the learning of $r_{\phi_{k+1}}(\mathbf{z})$. In practice, it is achieved by maintaining a memory matrix that stores a posterior samples $\tilde{\mathbf{z}}_i$ associated to each data point $\mathbf{x}_i$. Note that we only need to keep one memory matrix throughout the training, as only the posterior samples from the previous stage are needed.

**Test time sampling:** After obtaining the density ratio estimators in each stage and form the final EBM prior $p_\phi(\mathbf{z})$, we can sample latent variables $\mathbf{z} \sim p_\phi(\mathbf{z})$ and produce sample $\mathbf{x}$ by decoding $\mathbf{z}$. Sampling from $p_\phi(\mathbf{z})$ can be done by either running Langvin dynamics with $\nabla_{\mathbf{z}} \log p_\phi(\mathbf{z}) = \nabla_{\mathbf{z}} \left( \sum_{i=0}^{m-1} \log r_{\phi_i}(\mathbf{z}) - \frac{1}{2}\|\mathbf{z}\|^2 \right)$, or Sampling-Importance-Resampling (SIR) techniques.

## 4 Related Work

**Latent variable deep generative models:** Our proposed method aims to improve the performance of latent variable deep generative models. Such models consist of a decoder for generation, and require an inference mechanism to infer latent variables. VAEs [23, 46] learn the decoder network by training a tractable inference network (encoder) to approximate the intractable posterior distribution of the latent variables. Alternatively, Han et al. [13, 14], Xie et al. [52], Nijkamp et al. [33] infer the latent variables by Langevin sampling from the posterior distribution without using a encoder. Our method follows the latter approach that uses Langevin sampling to infer latent variables.

**Algorithm 1:** Adaptive Multi-stage Density Ratio Estimation.

---

**input** : Learning iterations $T$, number of stages $K$, observed examples $\{\mathbf{x}_i\}_{i=1}^n$, number of posterior
        sampling steps $L$, initial prior model $p_0(\mathbf{z})$
**output** : Estimated parameters $\theta, \phi_k, k = 1, \ldots, m$.
$k = 0$
**for** $t = 0 : T - 1$ **do**
    1. **Mini-batch**: Sample observed examples $\{\mathbf{x}_i\}_{i=1}^m$.
    2. **Posterior sampling for** $q_k(\mathbf{z})$ : For each $\mathbf{x}_i$, sample $\mathbf{z}_i \sim p_{\theta_t}(\mathbf{z}|\mathbf{x}_i)$ using Eq. (3) for $L$ steps with
       current prior $p_k(\mathbf{z})$.
    3. **Learning density ratio** $r_{\phi_k}(\mathbf{z})$: Update $\phi_k$ using contrastive estimation between $q_k(\mathbf{z})$ and $p_k(\mathbf{z})$
       via Eq. (4).
    4. **Learning generation model**: update $\theta$ according to Eq. (7)
    **if** *t is a multiple of* $T/K$ **then**
        5. **Stage transition and prior update**: Construct new stage $k = k + 1$, update the prior
        $p_k(\mathbf{z}) = r_{\phi_{k-1}}(\mathbf{z})p_{k-1}(\mathbf{z})$

---

**Discriminative contrastive estimation for learning generative models:** The efforts has been made to combine the discriminative and generative models [27, 20, 47], particularly, as introduced in Section 2.2, discriminative contrastive estimation can be applied to learning EBMs. Gao et al. [7] use a normalizing flow [38] as the base distribution for contrastive estimation. Aneja et al. [1] refine the prior distribution of a pre-trained VAE by noise contrastive estimation. However, such a method may fail if the empirical latent distribution (called aggregated posterior) is far away from the Gaussian noise. Rhodes et al. [42] propose telescoping density-ratio estimation, which breaks the estimation into several sub-problems. The method is connected to a range of methods leverage sequences of intermediate distributions such as [8, 29, 24].

**Generator model with flexible prior:** Our method trains an energy-based prior on the latent space by proposed adaptive multi-stage NCE, so our work is related to the broader line of previous papers on introducing flexible prior distribution. Tomczak and Welling [45] parameterized the prior based on the posterior inference model, and [2] proposed to construct priors using rejection sampling. Some previous work adopt a two-stage approach, which first trains a latent variable model with simple prior, and then trains a separate prior model to match the aggregated posterior distribution. For example, 2s-VAE [4] trains another VAE in the latent space; Ghosh et al. [9] fit a Gaussian mixture model on latent codes. Additional work in this line include [34, 5, 48, 50, 39, 53, 54].

Pang et al. [36] have the closest connection to our work. Similar to us, they introduce an EBM on the latent space. Both the latent space EBM and the generator network are learned jointly by maximum likelihood, and in particular the training involves short-run MCMC sampling from both the prior and posterior distributions. In contrast, we sequentially learn a more expressive EBM with our novel adaptive multi-stage NCE, which avoids running MCMC for EBM prior. We also show improved results on image generation and outlier detection tasks.

## 5 Experiments

In this section, we present a set of experiments which highlight the effectiveness of our proposed method. We want to show that our method can (i) learn an generator model with expressive prior distribution from which visually realistic images can be synthesized, (ii) generalize well by faithfully reconstructing test images during training, and (iii) successfully perform anomaly detection. To show the performance of our method, we mainly include SVHN [31], CelebA [28] and CIFAR-10 [25] in our study. Besides, we also include studies on the training dynamics and the Langevin sampling, as well as ablation studies to better understand our method. Details about the experiments, including network architecture, the choices of the model hyper-parameters and the optimization method for each dataset can be found in Appendix A.

### 5.1 Image Synthesis and Reconstruction

We evaluate the quality of the generated and reconstructed images. Ideally, if the model is well-trained, the EBM prior on latent space will fit the marginal distribution of latent variables, which in turn leads to realistic samples and faithful reconstructions. We benchmark our model against a variety

Table 1: MSE($\downarrow$) and FID($\downarrow$) obtained from models trained on different datasets. For our reported results, the FID is computed based on 50k generated images and 50k real images and the MSE is computed based on 10k test images.

| | SVHN | | CelebA | | CIFAR-10 | |
| --- | --- | --- | --- | --- | --- | --- |
| | MSE | FID | MSE | FID | MSE | FID |
| VAE [23] | 0.019 | 46.78 | 0.021 | 65.75 | 0.057 | 106.37 |
| ABP [13] | - | 49.71 | - | 51.50 | - | - |
| SRI [33] | 0.018 | 44.86 | 0.020 | 61.03 | - | - |
| SRI (L=5) [33] | 0.011 | 35.32 | 0.015 | 47.95 | - | - |
| 2s-VAE [4] | 0.019 | 42.81 | 0.021 | 44.40 | 0.056 | 72.90 |
| RAE [9] | 0.014 | 40.02 | 0.018 | 40.95 | 0.027 | 74.16 |
| NCP-VAE [1] | 0.020 | 33.23 | 0.021 | 42.07 | 0.054 | 78.06 |
| LEBM [36] | 0.008 | 29.44 | 0.013 | 37.87 | 0.020 | 70.15 |
| Adaptive CE (ours) | **0.004** | **26.19** | **0.009** | **35.38** | **0.008** | **65.01** |

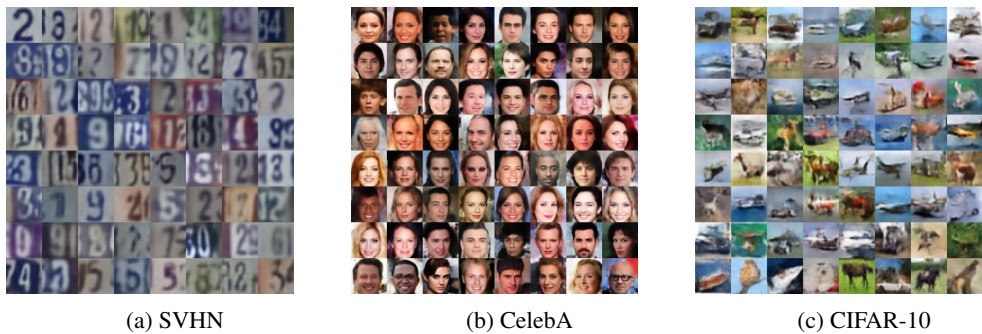

| (a) SVHN | (b) CelebA | (c) CIFAR-10 |

Figure 2: Samples generated from our models trained on SVHN, CelebA and CIFAR-10 datasets.

of previous methods including VAE [23], Alternating Back-propogation (ABP) [13] and Short-run Inference (SRI) [33] which assume a simple standard Gaussian prior distribution for the latent vector, as well as recent two-stage methods such as 2-stage VAE [4], RAE [9] and NCP-VAE [1], whose prior distributions are learned with posterior samples in a second stage after the generator is trained. We also compare our method with LEBM [36], which learns a EBM prior adaptively during training the generator, while the EBM prior is trained by maximum likelihood instead of density ratio estimation. To make fair comparisons, we follow the protocol as in [36].

**Synthesis:** We report the quantitative results of FID [16] in Table 1, where we observe that across all datasets, our proposed method achieves superior generation performance compared to baseline models based with simple or learned prior distribution.

We show qualitative results of generated samples in Figure 2, where we observe that our model can generate diverse, sharp and high-quality samples. Additional qualitative samples are presented in Appendix C. To test our method's scalability, we trained a larger generator on CelebA-HQ ($128 \times 128$) and show samples in Appendix C, the model can produce realistic samples.

**Reconstruction:** Note that the posterior Langevin dynamics should not only help to learn the latent space EBM prior model but also produce samples that approximately come from true posterior distribution $p_\theta(\mathbf{z}|\mathbf{x})$ of the generator model. To verify this, we evaluate the accuracy of the posterior inference by looking at reconstruction error on test images. We quantitatively compare reconstructions of test images with baseline models using mean square error (MSE) in Table 1. We observe that our method consistently obtain lower reconstruction error than competing methods do. We also provide qualitative results of reconstruction in Appendix B.

## 5.2 Anomaly Detection

Anomaly detection is another task to evaluate the generator model. With a generator and an EBM prior model trained on in-distribution data, the posterior $p_\theta(\mathbf{z}|\mathbf{x})$ would have separated probability

Table 2: AUPRC($\uparrow$) scores for unsupervised anomaly detection on MNIST. Numbers are taken from [36] and results for our model are averaged over last 10 trials to account for variance.

| Heldout Digit | 1 | 4 | 5 | 7 | 9 |
|---|---|---|---|---|---|
| VAE [23] | 0.063 | 0.337 | 0.325 | 0.148 | 0.104 |
| ABP [13] | $0.095 \pm 0.03$ | $0.138 \pm 0.04$ | $0.147 \pm 0.03$ | $0.138 \pm 0.02$ | $0.102 \pm 0.03$ |
| MEG [26] | $0.281 \pm 0.04$ | $0.401 \pm 0.06$ | $0.402 \pm 0.06$ | $0.290 \pm 0.04$ | $0.342 \pm 0.03$ |
| BiGAN-$\sigma$ [55] | $0.287 \pm 0.02$ | $0.443 \pm 0.03$ | $0.514 \pm 0.03$ | $0.347 \pm 0.02$ | $0.307 \pm 0.03$ |
| LEBM [36] | $0.336 \pm 0.01$ | $0.630 \pm 0.02$ | $0.619 \pm 0.01$ | $0.463 \pm 0.01$ | $0.413 \pm 0.01$ |
| Adaptive CE (ours) | $\mathbf{0.531 \pm 0.02}$ | $\mathbf{0.729 \pm 0.02}$ | $\mathbf{0.742 \pm 0.01}$ | $\mathbf{0.620 \pm 0.02}$ | $\mathbf{0.499 \pm 0.01}$ |

densities for in-distribution and out-of-distribution (anomalous) samples. In particular, we decide whether a test sample $\mathbf{x}$ is anomalous or not by first sampling $\mathbf{z}$ from the posterior $p_\theta(\mathbf{z}|\mathbf{x})$ by short-run Langevin dynamics, and then computing the joint density $p_{\theta,\phi}(\mathbf{x}, \mathbf{z}) = p_\theta(\mathbf{x}|\mathbf{z})p_\phi(\mathbf{z})$. A higher value of log joint density indicates the test sample is more likely to be a normal sample. Some prior work on using latent variable generative model for anomaly detection includes [49, 15, 36].

Following the experimental settings in [26, 55], we set each class in the MNIST dataset as an anomalous class and leave the other 9 classes as normal. Note that it is a challenging task and all previous methods do not perform well. To evaluate the performance, we use the log-posterior density to compute the area under the precision-recall curve (AUPRC) [6]. We compare our method with related models in Table 2, where we observe that our method obtains significant improvements.

## 5.3 Analyzing Training Loss

In Figure 4, we plot the evolution of the density ratio estimation loss (Eq. 4) for each stage of estimation during training. Our experiment has 4 estimation stages, resulted in 4 density ratio estimators. We observe that the loss for the first stage, which estimates the density ratio between unit Gaussian prior $p_0(\mathbf{z})$ and aggregated posterior is significantly lower than later stages, which estimate the ratio between the updated prior and updated posterior. This observation is consistent with our intuition: directly discriminating between Gaussian prior and posterior is very easy, while introducing additional stages of estimation make the task more difficult, and hence the estimated density ratio is more reliable. Higher losses in later stages also suggests that the prior is getting close to aggregated posterior, as the discrimination becomes harder.

## 5.4 Analyzing Langevin Dynamics

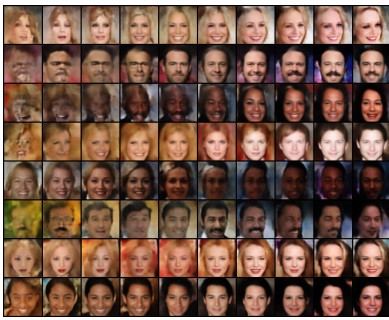

Figure 3: Transition of Langevin dynamics initialized from $p_0(\mathbf{z})$ towards $p_\phi(\mathbf{z})$ for 200 steps.

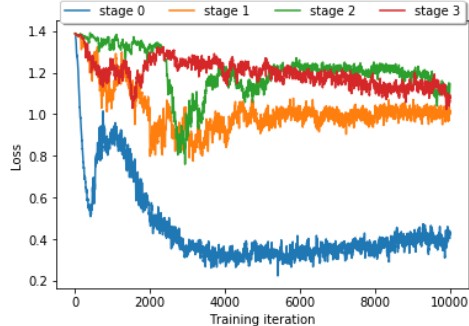

Figure 4: Density ratio estimation loss for each estimation stage.

In Figure 3, we visualize the transition of Langevin dynamics initialized from $p_0(\mathbf{z})$ towards $p_\phi(\mathbf{z})$ on a model trained on CelebA. The LD iterates for 200 steps, which is longer than the LD for training (30 steps). We expect that with a well-trained $p_\phi(\mathbf{z})$, the trajectory of a Markov chain should transit towards samples of higher quality. Indeed, we observe that the quality of synthesis improves significantly with as the LD progresses. In addition, we observe human faces with different

Table 3: Results for ablation study on CelebA dataset.

| Method | Adaptive | | | | | Non-adaptive | | | | |
|---|---|---|---|---|---|---|---|---|---|---|
| # of stages | 0 | 1 | 2 | 4 | 8 | 0 | 1 | 2 | 4 | 8 |
| FID | 62.78 | 44.17 | 39.85 | **35.38** | 35.84 | 62.78 | 43.84 | 42.61 | 42.48 | 43.06 |

identities along the LD, suggesting that the Markov chain can mix between different modes of the prior distribution. This indicates that the density function of learned EBM prior has a smooth geometry that allows MCMC to mix well.

## 5.5 Ablation Study

To better understand our proposed method, we conduct ablation study on number of density ratio estimators and training methods. We use CelebA for the ablation experiments.

**Number of stages.** The most important hyper-parameter of our method is the number of density ratio estimators, or equivalently, the number of training stages. We present the FID score of models trained with different number of stages in the first part of Table 3. The line of 0 stage means no latent EBM at all, i.e., simply training a generator model by short-run inference and sampling from it by decoding $\mathbf{z} \sim p_0(\mathbf{z})$.

We make the following observations. Firstly, directly sampling latent variables from $p_0$ leads to poor results, while any latent EBM trained by density ratio estimation significantly improves the performance, suggesting the necessity of learning the latent EBM. Secondly, we see that multi-stage density ratio estimation can further significantly improve the performance of single-stage estimation. The results indicate that multi-stage density ratio estimation facilitates the training of latent EBM by gradually making the estimation task harder. We observe that the FID score does not improve for more than 4 stages, and therefore we choose 4 as the number of stages for our main experiments.

**Training method: adaptive vs. non-adaptive.** It is important to distinguish our method from TRE in Rhodes et al. [42]. TRE assumes the target distribution to be fixed, therefore, if we adopt TRE, the posterior distribution $p_\theta(\mathbf{z}|\mathbf{x})$ will be a fixed one throughout the training. In contrast, our training method is adaptive in the sense that the target posterior is updated by incorporating the current EBM prior into the joint distribution when a new stage is introduced. To quantitatively compare these two approaches, we also train non-adaptive version of the model and report the numbers in the second part of Table 3. We observe that models trained with non-adaptive multi-stage density ratio estimation obtain significantly worse results. Therefore, we believe that it is crucial to learn density ratios sequentially with adaptive posterior.

## 5.6 Parameter Efficiency

One potential disadvantage of our method is its parameter inefficiency from multiple estimator networks. Moreover, since the training is sequential, we cannot share parameters between estimators as done in Rhodes et al. [42]. Fortunately, our EBM is on the latent space so the network is light-weighted. For example, with 4 density ratio estimators, the number of parameters in the prior EBM is only around 1% of the number of parameters in the generator. In addition, we confirm the larger number of parameters in the latent EBM is not the cause of improvements, as we train a single stage model with 4× size and observe no improvement.

## 6 Conclusions

In this paper, we propose adaptive multi-stage density ratio estimation, which is an effective method for learning a EBM prior for a generator model. Our method learns the latent EBMs by introducing multiple density ratio estimators that learn the density ratio between prior and posterior sequentially and adaptively. We demonstrate the effectiveness of our method by conducting comprehensive experiments, and empirical results show the advantage of our method on generation, reconstruction and anomaly detection tasks.

As future directions, our method can potentially be applied to modeling the latent space of generator models in other domains, such as text [35] and graph. We also tend to develop more advanced and efficient inference schemes for posterior density estimation.

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
