# A Experimental Details

In this section, we introduce the detailed settings of our experiments.

## A.1 Datasets

We mainly study our method with SVHN [31] ($32 \times 32 \times 3$), CIFAR-10 [25] ($32 \times 32 \times 3$), and CelebA [28] ($64 \times 64 \times 3$). Following Pang et al. [36], we use the full training set of SVHN ($73,257$) and CIFAR-10 ($50,000$), and take $40,000$ examples of CelebA as training data following [32]. The training images are resized and scaled to $[-1, 1]$.

## A.2 Network architectures.

For experiments on SVHN, CelebA and CIFAR-10, each density ratio estimator network has a simple fully-connected structure described in Table 4.

Table 4: Network structures for density ratio estimator. LReLU indicates the Leaky ReLU activation function. The slope in Leaky ReLU is set to be 0.1.

| Layers | In-Out Size |
|---|---|
| Input: $z$ | 100 |
| Linear, LReLU | 200 |
| Linear, LReLU | 200 |
| Linear | 1 |

We let the generator network having a simple deconvolution structure, similar to DCGAN [40]. The generator network for each dataset is depicted in Table 5.

Table 5: Network structures for the generator networks of SVHN, CelebA, CIFAR-10 (from top to bottom). convT($n$) indicates a transposed convolutional operation with $n$ output channels. LReLU indicates the Leaky-ReLU activation function. The slope in Leaky ReLU is set to be 0.2.

| Layers | In-Out Size | Stride |
|---|---|---|
| Input: $x$ | 1x1x100 | - |
| 4x4 convT(ngf x 8), LReLU | 4x4x(ngf x 8) | 1 |
| 4x4 convT(ngf x 4), LReLU | 8x8x(ngf x 4) | 2 |
| 4x4 convT(ngf x 2), LReLU | 16x16x(ngf x 2) | 2 |
| 4x4 convT(3), Tanh | 32x32x3 | 2 |
| Layers | In-Out Size | Stride |
| Input: $x$ | 1x1x100 | - |
| 4x4 convT(ngf x 8), LReLU | 4x4x(ngf x 8) | 1 |
| 4x4 convT(ngf x 4), LReLU | 8x8x(ngf x 4) | 2 |
| 4x4 convT(ngf x 2), LReLU | 16x16x(ngf x 2) | 2 |
| 4x4 convT(ngf x 1), LReLU | 32x32x(ngf x 1) | 2 |
| 4x4 convT(3), Tanh | 64x64x3 | 2 |
| Layers | In-Out Size | Stride |
| Input: $x$ | 1x1x128 | - |
| 8x8 convT(ngf x 8), LReLU | 8x8x(ngf x 8) | 1 |
| 4x4 convT(ngf x 4), LReLU | 16x16x(ngf x 4) | 2 |
| 4x4 convT(ngf x 2), LReLU | 32x32x(ngf x 2) | 2 |
| 3x3 convT(3), Tanh | 32x32x3 | 1 |

## A.3 Training Hyper-parameters

We introduce some hyper-parameter setting for training our model. For the main experiments, we have 4 density ratio estimation stages. We adopt the persistent approach for generating samples from prior distribution. For the posterior sampling Langevin dynamics, we use step size $0.1$, and run the LD for 30 steps for SVHN and CelebA, and 40 steps on CIFAR-10.

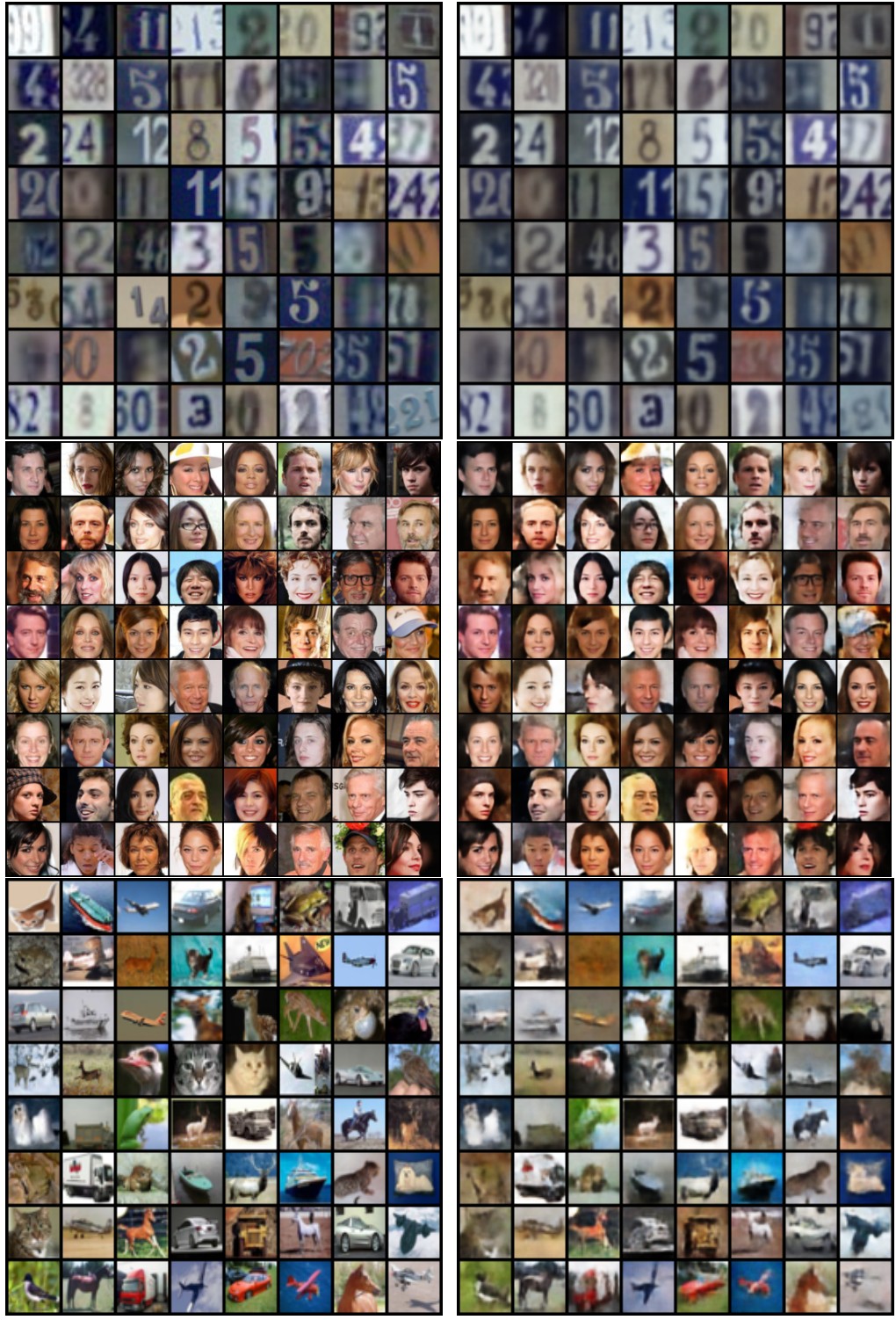

Figure 5: Qualitative results of reconstruction on test images. Left: real images from test set. Right: reconstructed images by sampling from the posterior.

The parameters for the density ratio estimators and image generators are initialized with Xavier initialization [10]. We train both the generator and density ratio estimators using Adam [22] optimizer. The learning rate for the generator is $1e-4$ and the learning rate for the density ratio estimator is

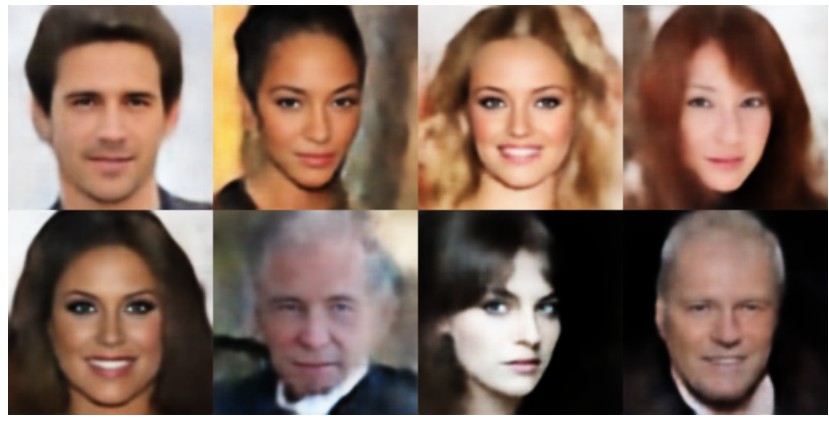

Figure 6: High-resolution samples on CelebA-HQ.

$5e - 5$. We train the model for 100 epochs for SVHN and CelebA, where a new estimation stage is introduced every 25 epochs. For CIFAR-10, we train the model for 200 epochs for SVHN and CelebA, where a new estimation stage is introduced every 50 epochs.

During test stage, we run LD on the learned EBM prior with step size 0.1 for 100 steps.

## B  Reconstruction Samples

In Figure 5, we provide some qualitative examples of reconstructing test images. We see that our model can reconstruct unseen images faithfully.

## C  Additional Qualitative Results

We trained our model on high-resolution $128 \times 128$ CelebA-HQ, samples are shown in Figure 6 We also provide additional qualitative samples from our models trained on SVHN, CelebA and CIFAR-10 in Figure 7.

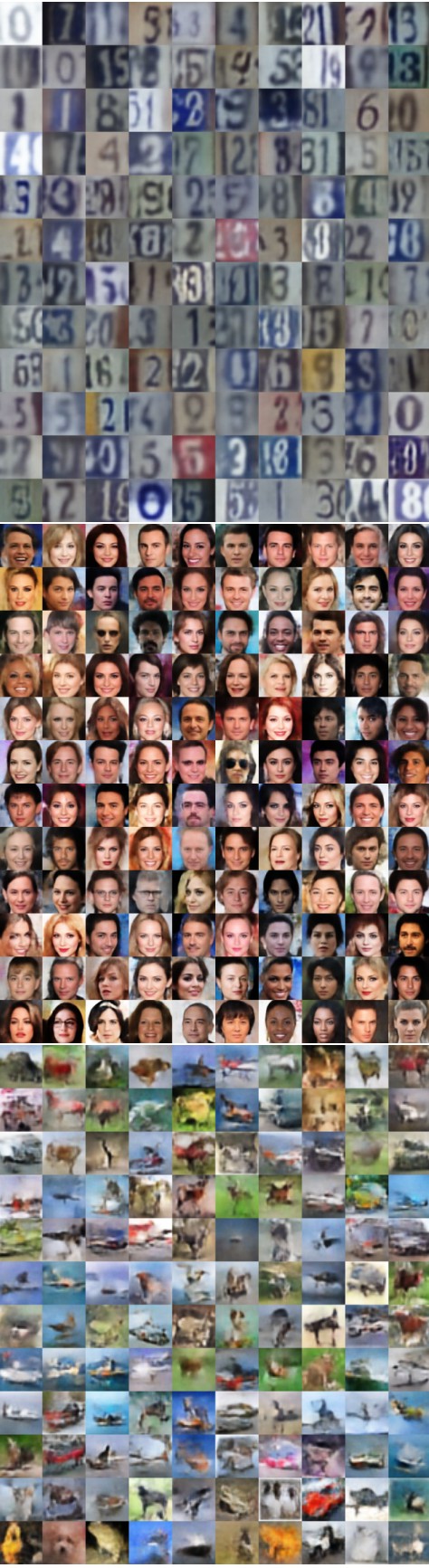

Figure 7: Additional randomly generated samples from our models.