# OpenReview forum: "Adaptive Multi-stage Density Ratio Estimation for Learning Latent Space Energy-based Model"
_NeurIPS.cc/2022/Conference — NeurIPS 2022 Accept_

### Official Review · Reviewer_JvzX · 2022-07-11

**Rating:** 7
**Confidence:** 4
**Soundness:** 3 good
**Presentation:** 3 good
**Contribution:** 3 good

**Summary:**

This paper proposes using EBMs in the latent space before being pushed through a latent variable model. The paper proposes learning this EBM using NCE instead of via MCMC sampling. The difference between the prior and the latent posterior is estimated using short-run langevin instead of a recognition network (variational inference). This technique is applied repeatedly to obtain several “stages” of density ratio estimation.

The paper demonstrates their approach on image generation and reconstruction, as well as on anomaly detection.


**Questions:**

Based on figure 4, it sounds like there are decreasing marginal returns to using additional stages. Is this a correct assessment? How do you propose to choose the number of stages?


**Limitations:**

I see no potential for negative societal impact.

**Strengths And Weaknesses:**

The experiment in Section 5.4 on analyzing langevin dynamics is very interesting. It uniquely shows the benefit of using a latent variable model, with the prior learned by an EBM, and how this results in stable, well-mixed markov chains.

---

> ### Author Response · Authors · 2022-08-02
> **Response to Reviewer JvzX**
>
> We thank the reviewer for the positive feedback. Here we will try our best to answer the questions.
>
> **Decreasing marginal returns to using additional stages**:
>
> Yes, we do observe that the marginal returns of using additional stages decrease. Note that at each stage, the training of the density ratio estimator corresponds to distinguishing between posterior samples and prior samples, and our novel adaptive prior set-up will make the prior closer to the posterior when a new stage is added. This will make the classification between posterior and prior more difficult (see figure 4 in our submission, where the classification loss is larger for latter stages). When the newly introduced density estimator does not have enough capacity to classify between posterior and prior, the learning is saturated and hence the newly added stage cannot further improve the performance. Currently we use the same model structure for the density estimator for each stage. We may explore gradually increasing the capacity of the density estimator in new stages, as the classification problem becomes harder as training progresses. However, doing so will introduce more computational overhead.
>
> The proper number of stages varies with data and model design, currently we choose the number of stages by monitoring the training loss. When the classification becomes very difficult (as indicated by the loss function ), we may stop adding more stages.

---

> > ### Comment · Reviewer_JvzX · 2022-08-04
> > **Response**
> >
> > Thank you for your response and your detailed answer to my question. This clears up the uncertainty I had, and I am happy to increase my score.

---

### Official Review · Reviewer_o7M9 · 2022-07-18

**Rating:** 7
**Confidence:** 4
**Soundness:** 3 good
**Presentation:** 3 good
**Contribution:** 3 good

**Summary:**

The authors propose a new method for unsupervised learning of latent models that consist of an energy based prior model (EBPM) and a (stochastic) deep decoder.  For the EBPM estimation, the authors propose to extend the approach of Telescoping density-Ratio Estimation (TRE) (Rhodes et al 2020).  The utility of the approach is evaluated on several image modeling benchmarks (SHVN, CelebA /CelebA-HD, CIFAR-10) and on an outlier detection task on MNIST.

**Questions:**

* what is the tradeoff between the increased complexity of EBPM vs a more complex decoder?  Current generative SoTA for CelebA-HD is certainly not based on the simple decoder model architectures used here (c.f., StyleGAN type of models).
* Please comment on the increase in computational complexity of learning (time-complexity) compared to eg simple Vamp prior


**Limitations:**

Yes

**Strengths And Weaknesses:**

Strengths:
+ The proposed approach is clearly described, well-motivated, and appears technically correct
+ While the approach uses the same telescoping/recursive approximation idea as TRE, the idea is applied in a sufficiently novel manner to EBPM + generative model estimation
+ All results point to clear benefits of the proposed approach; competing approaches constitute a reasonable selection of SoTAs for unsupervised latent model estimation + most related approach (LEBM)
+ authors provide insightful analyses of ablation wrt #stages and ratio estimation "hardness"

Weaknesses:
- while I do not see any major downsides to the paper, it would be useful to comment on the increased complexity (during learning and inference) introduced by the telescoping ratio-based EBPM

---

> ### Author Response · Authors · 2022-08-02
> **Response to Reviewer o7M9**
>
> We thank the reviewer for the positive feedback. Here we will try our best to answer the questions.
>
> **Comment on the increased complexity (during learning and inference) introduced by the telescoping ratio-based EBPM**:
>
> In general, we think the extra computational overhead introduced by adaptive multi-stage NCE is very small. First of all, the training of our model uses a memory bank to store the posterior samples in the previous stage as approximation to the prior samples in the current stage, which means that unlike [1], we do not need to run expensive MCMC to sample from the prior during training. This makes our model even more computationally efficient than existing EBPM. Secondly, although we increase the size of the prior model by introducing a separate model for each stage, we want to emphasize that the prior models to perform density ratio matching are really light-weighted. They are small MLPs with a few layers, which would not introduce much overhead in training or sampling. The computational efficiency of our model is discussed in section 5.6.
>
> **Compare efficiency to Vamp Prior**:
>
> We thank the reviewer for pointing out this interesting comparison. Our model shares the high-level goal with Vamp-prior VAE, which is to design a better prior model to approximate the aggregated posterior. However, Vamp-prior VAE tackles this issue by introducing a mixture of variational posteriors with pseudo-inputs as a new prior model for the VAE. It belongs to the VAE model as it uses an additional inference (encoder) model to obtain posterior samples, in contrast, our model is encoder-free and adopts the expressive EBM as the prior.
>
> For computational efficiency, in general, VAEs are easier to train and require less computation, because they do not need to run iterative MCMC for posterior sampling during training. However, the proposed EBM prior model in the paper offers better likelihood modeling (reflected in test reconstruction) and sample quality. Given the same decoder size, latent EBM models typically perform significantly better in reconstruction and sample quality than VAE-based models.
>
> Finally, vamp-prior involves optimizing the pseudo inputs, which also introduces some computational overhead.
>
> **Tradeoff between the increased complexity of EBPM vs a more complex decoder**:
>
> We would like to thank the reviewer for bringing this interesting question. While it is interesting to explore more advanced decoder structures, which will serve as our future study, we focus on designing the prior model in this paper and hence evaluate our model in a standard protocol following related works.
>
> StyleGAN makes some interesting modifications to the generator structure compared to other GANs. However, its adversarial training objective also plays a crucial part for high-quality generation. For fair comparison,  in this paper, we only  focus on the non-adversarial, likelihood-based models. Models such as NVAE[2] use large decoders, but more importantly, they carefully design the probabilistic model of their latent variables. As observed in [2], a large decoder alone would not guarantee the strong model performance. Therefore, we believe that for latent variable likelihood-based models, it is more effective to design the probabilistic model (such as designing a prior model as we do) than to increase the complexity of the decoder.
>
> Finally, we would like to emphasize that our model only introduces a negligible amount of parameters in the prior, compared to the parameter size of the decoder. The number of parameters in the prior is less than 1% of the number of parameters in the decoder.  Therefore, while we will try those advanced structures for better performance, but we also point out that the model size (# of parameters) will increase dramatically. Given limited “budget”, it would be helpful to design the informative prior instead.
>
> [1] Learning Latent Space Energy-Based Prior Model, Pang et al.
>
> [2] NVAE: A Deep Hierarchical Variational Autoencoder, Vahdat and Kautz.

---

### Official Review · Reviewer_PebL · 2022-08-02

**Rating:** 7
**Confidence:** 4
**Soundness:** 3 good
**Presentation:** 3 good
**Contribution:** 2 fair

**Summary:**

This paper proposes a new adaptive multi-stage density ratio estimation framework for learning latent space energy-based model (EBM) in order to overcome the density-chasm problem. Namely, whenever the gap between p (prior distribution) and q (posterior data distribution), the classifier can obtain almost perfect accuracy with a relative poor estimate of the density ratio. The adaptive multi-stage density ratio estimation follow the telescope density-ratio estimation (TRE) approach, learns the latent space EBM through multiple stages. They claim that their work shares similar root as the Product-of-Expert (PoE), each intermediate ratio is treated as individual expert model and it has the potential to produce much sharper distribution than a single model. In training their proposed work, instead of using MLE, they resort to
multi-stage NCE for obtaining prior models (also avoid using expensive MCMC sampling) and short-run Langevin Dynamics (LD) to infer the posterior. Their empirical results show the effectiveness of the proposed work. They also do some ablation studies to show how to determine stage numbers and the importance of adaptiveness.

**Questions:**

1. Following the sequential nature of this work, we wonder if this work is really much more efficient than MCMC.
2. The selection of stage number as four based on FID scores is not so convincing. As we mentioned before, this lacks theoretical guarantee and looks not generic either.
3. Further, they still resort to short-run LD for posterior inferences while current standard EBM usually also make use of short-run LD for inferences. So, one question we have is when the number of stages is quite many, can is it more efficient than the standard EBM models,

**Limitations:**

1. Please add theoretical analysis or proofs on the stage choices or sample efficiencies if possible
2. Looks that the constraint of adaptive to sequential learning is not convincing. Is it possible to combine sequential and distributional?
3. They did not run experiments with TRE, which is the most closely related work. So, we do not have ideas whether their work can really work better than related works.

**Strengths And Weaknesses:**

Strengths:
1. The idea of learning EBM via multi-stage density estimation is somewhat novel and also workable for overcoming the density-chasm problem.
2. This work has done intensive ablation studies to show the importance of stage numbers and whether to use adaptive or non-adaptive.
3. They have done experiments in SVHN, CelebA and CIFAR-10 to demonstrate the effectiveness of their proposal.
4. They also provide the workable code and this make reproduction and extensions more feasible.
Weaknesses:
1. This work can still be regarded as an incremental research from the Telescoping density-ratio estimation.
2. Compared to TRE, it seems that it is much less efficient since each divided component of TRE is done simultaneously and can be trained in a distributional fashion while this work is done sequentially and later stage depends on previous stages.
3. This work lacks theoretical analysis and proofs to understand the effectiveness and efficiency of proposed approaches.

---

> ### Author Response · Authors · 2022-08-03
> **Response to Reviewer PebL**
>
> We thank the reviewer for the positive feedback. Here we will try our best to address the concerns and answer the questions. Below, Q refers to Questions, L refers to Limitations.
>
> **Q1: Efficiency compared to standard EBM Models**
>
> We argue that our model is more efficient than standard EBM models. Standard EBM models directly learn the EBM on pixel space, while our model learns the EBM on latent space. MCMC sampling on pixel space is less efficient and hardER to mix, since the dimension is very high (e.g., ~ 3000 dims for a small 32x32x3 image) and the pixel distribution is highly multimodal and complicated, therefore more MCMC steps are required. In contrast, in latent space EBM, we sample a lower-dimensional latent (e.g., ~ 100 dims ) variable on a simpler and less multimodal distribution (with Gaussian as the base distribution), so posterior sampling from latent EBM is simpler and requires less computation.
>
> Our model is an improvement to the standard latent EBM model such as [1]. The posterior inference of our model only adds negligible computational overhead compared to standard latent EBM model, because the prior networks (consists of several small MLPl density ratio estimators) are very light-weighted (less than 1% of the decoder).
>
> **Q2 and L1: Selection of stage number and theoretical explanation**
>
> Learning an EBM prior with density ratio estimation can be seen as a classification problem to distinguish between prior and posterior samples, and what we do is to make the classification problem increasingly difficult (as too easy classification will not produce meaningful density ratio estimation). As a result, besides looking at the FID, we can also look at the classification loss (presented in Figure 4). When the loss does not decrease (which means that the classification becomes random guessing), there is no benefit of adding more stages.
>
> We appreciate the reviewer for pointing out the interesting question of theoretical analysis of choosing the number of stages. We believe that the ideal number of stages largely depends on the complexity of the dataset and the capacity of the density ratio estimators.As our future work, we will thoroughly explore the theoretical insights of the stage selection as you suggested.
>
>
> **Q3: Efficiency compared to MCMC**
>
> Our model trains the latent EBM prior with density ratio estimation. We argue that compared to [1] that learns the EBM prior with maximum likelihood (which uses MCMC sampling), our model is more efficient to train as we do not require MCMC sampling from prior during training. The efficiency is not harmed by the multi-stage, sequential training for the two reasons:
>
> (1) we train the model with the same total number of iterations as the latent EBM model [1] trained by maximum likelihood, and our model performance is better (Table 1 in paper). This means that our model does not take longer training iterations for better performance.
>
> (2) For each training iteration, our model is more efficient as we do not need to run the prior MCMC sampling which can be costly. However, we admit that the posterior MCMC is indeed needed for better inference, and this is shared by [1] and our model.
>
>
> **L3: Run experiments with TRE**
>
> The “non-adaptive” case in Table 3 corresponds to directly applying TRE in latent EBM training, where we observe that our model performs better than naively applying TRE. TRE is non-adaptive, in the sense that it aims to estimate the density ratio between two fixed distributions. In contrast, our target distribution (which is the posterior) is updated by incorporating the current EBM prior into the joint distribution when a new stage is introduced. Note that as the title suggests, our model has two highlights: multi-stage and adaptive, and the adaptive part is what distinguishes us from TRE. More details are provided in section 5.5.
>
> We apologize for the confusion. We should explicitly mention in the caption of Table 3 that the non-adaptive case corresponds to TRE, and we will make it clear in the final revision.
>
>
> **L2: Is it possible to combine sequential and distributional?**
>
> If we understand correctly, you are asking whether our model can be trained distributedly. You are correct that the training is sequential, which means that the training of the second stage of density ratio estimation requires the already trained density ratio estimator in the first stage. However, if we have multiple machines and want to accelerate the training, we can simply allocate the training batch into different machines, and that would lead to a significant speed-up. Besides, as observed empirically, the number of stages needed can be small. Your suggestion about combining sequential and distributional learning surely offers an interesting research direction.
>
> [1] Learning Latent Space Energy-Based Prior Model, Pang et al.

---

> > ### Author Response · Authors · 2022-08-03
> > **Thank you for the review!**
> >
> > Please let us know if there’s any misunderstanding regarding your feedback. Since this is an emergency review, we have very limited time to respond. We really appreciate your efforts to submit an emergency review, thank you!

---

> > > ### Comment · Reviewer_PebL · 2022-08-07
> > > **Response**
> > >
> > > Thank you for your response and your detailed answer to my question. This clears up the uncertainty I had, and I am happy to increase my score.

---

### Public Comment · ~Qianjun_Li1 · 2023-02-27
**Impressive Work!**

Your work is great, I just wanna know is there any code for reproducing your result in the paper?

---

### Meta-Review · Area_Chair_BVzv · 2022-08-27

**Recommendation:** Accept
**Confidence:** Certain

**Metareview:**

Reviewers unanimously agree that this submission is of good technical quality and well-written. The author proposed an unsupervised learning paradigm for latent variable models, based on EBMs and NCE, by extending the prior work on telescoping density-ratio estimation


This paper proposes using EBMs in the latent space before being pushed through a latent variable model. The paper proposes learning this EBM using NCE instead of via MCMC sampling. The difference between the prior and the latent posterior is estimated using short-run langevin instead of a recognition network (variational inference). This technique is applied repeatedly to obtain several “stages” of density ratio estimation. Reviewers find the adaptive multi-stage method interesting and are convinced that it is effective. Authors rebuttal also helped Reviewers' understanding on this paper.


**Award:**

No

---

### Decision · Program_Chairs · 2022-09-14

Accept